# Management of Early-Stage Cervical Cancer: A Literature Review

**DOI:** 10.3390/cancers14030575

**Published:** 2022-01-24

**Authors:** Yasmin Medeiros Guimarães, Luani Rezende Godoy, Adhemar Longatto-Filho, Ricardo dos Reis

**Affiliations:** 1Molecular Oncology Research Center, Barretos Cancer Hospital, São Paulo 14784-400, Brazil; yasminmguimaraes@outlook.com (Y.M.G.); luanirgodoy@gmail.com (L.R.G.); longatto@med.uminho.pt (A.L.-F.); 2Medical Laboratory of Medical Investigation (LIM) 14, Department of Pathology, Medical School, University of São Paulo, São Paulo 01246-903, Brazil; 3Life and Health Sciences Research Institute (ICVS), School of Medicine, University of Minho, 4710-057 Braga, Portugal; 4ICVS/3B’s—PT Government Associate Laboratory, 4710-057 Braga, Portugal; 5ICVS/3B’s—PT Government Associate Laboratory, 4805-017 Guimarães, Portugal; 6Department of Gynecologic Oncology, Barretos Cancer Hospital, São Paulo 14784-400, Brazil

**Keywords:** cervical cancer, diagnosis and staging, radical surgery, sentinel lymph node, fertility sparing

## Abstract

**Simple Summary:**

Despite being a preventable disease, cervical cancer still causes morbidity and deaths worldwide. In the early stages (FIGO IA1 with lymph-vascular space invasion-IIA1), the disease is highly curable. The primary treatment for early-stage cervical cancer is radical hysterectomy with pelvic lymphadenectomy. This surgical treatment has changed during the past decades, and we aimed to review and discuss the advances in the literature. We performed a literature review through PubMed focusing on English articles about the topic of surgical management of early-stage cervical cancer. The emergent topics considered here are the FIGO 2018 staging system update, conservative management for selected patients, sentinel lymph node mapping, fertility preservation, surgical approach, and management of tumors up to 2 cm. These topics show an evolvement to a more tailored treatment to prevent morbidity and assure oncologic safety.

**Abstract:**

Cervical cancer (CC) remains a public health issue worldwide despite preventive measures. Surgical treatment in the early-stage CC has evolved during the last decades. Our aim was to review the advances in the literature and summarize the ongoing studies on this topic. To this end, we conducted a literature review through PubMed focusing on English-language articles on the surgical management of early-stage CC. The emergent topics considered here are the FIGO 2018 staging system update, conservative management with less radical procedures for selected patients, lymph node staging, fertility preservation, preferred surgical approach, management of tumors up to 2 cm, and prognosis. In terms of updating FIGO, we highlight the inclusion of lymph node status on staging and the possibility of imaging. Regarding the preferred surgical approach, we emphasize the LACC trial impact worldwide in favor of open surgery; however, we discuss the controversial application of this for tumors < 2 cm. In summary, all topics show a tendency to provide patients with tailored treatment that avoids morbidity while maintaining oncologic safety, which is already possible in high-income countries. We believe that efforts should focus on making this a reality for low-income countries as well.

## 1. Introduction

Cervical cancer (CC) remains an important cause of morbidity and mortality worldwide. The necessary cause of CC development is the persistent infection by high-risk human papillomavirus (HPV) [1]. After sexual transmission, HPV can reach the basal cells from the epithelium through microlesions and induce carcinogenesis over the years [1,2]. Given that this tumor is related to virus infection, especially HPV types 16 and 18, it is largely preventable too. The primary prevention is the vaccination against HPV [3] and secondary prevention consists of population-based screening. The above-mentioned differences in rates between high and low-income countries are mainly the consequences of the effectiveness of these prevention strategies. Among low-income countries vaccination is scarce and the screening is based on opportunistic cervical cytology programs, while in high-income countries, the vaccination has good coverage, and the DNA-HPV test is suitable for screening. The American Cancer Society, the Society of Gynecologic Oncology, and the European Guidelines for Quality Assurance in Cervical Cancer Screening recommend the DNA-HPV test as the best strategy for primary screening [4,5,6,7].

Despite these prevention measures, in 2020, about 604,000 new cases and 342,000 deaths were deported [8], representing the fourth most common cancer among women around the world [9]. The American Cancer Society estimates 14,480 new cases of CC and 4290 deaths by this tumor among women in the USA in 2021 [10]. In low-income countries, the situation is even more critical, as CC represents the second most common type of cancer (15.7 per 100,000 females) and the third most common cause of cancer mortality (8.3 per 100,000) [8,11]. This scenario indicates the relevance of the discussions and advances in CC treatment, especially in the early stages in which cure rates are high when well treated [12]. 

Currently, radical hysterectomy (RH) with bilateral pelvic lymphadenectomy has been considered the primary treatment for patients with CC in early stages (FIGO stages IA1 with lymph-vascular space invasion (LVSI), IA2, IB1, IB2, and IIA1) [13]. The RH was first described and published in 1912 by Wertheim (Austria, 1864–1920). In 1944, Meigs (USA, 1892–1963) contributed with the bilateral pelvic lymphadenectomy technique [14]. Then, the procedure known today as RH was called Wertheim-Meigs surgery, which includes the removal of the uterus, vagina, and parametrium [14]. Furthermore, the bilateral pelvic lymphadenectomy is conducted with the removal of four major groups of lymph nodes: ureteral, hypogastric, obturator, and iliac [15]. 

Although RH is established as the standard care for patients with CC in early stages, it is important to notice the recommendations and peculiarities regarding the diagnostic and accurate staging, lymph node evaluation, fertility-sparing options, preferred surgical approach, possibility of more conservative surgery in specific groups of patients, and finally the differences between what is suitable for low-income and high-income countries. Considering these emergent points in the literature in recent years, we aimed to provide a concise overview of the recommendations and advances in early-stage CC treatment.

## 2. Diagnosis

The clinical presentation of CC is characterized by post-coital or abnormal vaginal bleeding [16], rarely with malodorous vaginal discharge [17]. However, in the early stages, it is commonly asymptomatic, being diagnosed by screening or pelvic examination for other reasons [16]. The gold-standard method for CC diagnosis is the histopathological examination of a cervical biopsy: direct biopsy or a cone biopsy (conization) or endocervical curettage for endocervical lesions. The triad of clinical assessment, cytology, and colposcopy are the paths to reach it. The chosen means of initiating the investigation depends on the patient’s presentation and the availability of resources; nonetheless, clinical evaluation is the preferable starting point.

In clinical assessment, if a lesion is visible, or in the presence of an irregular, endured, or vegetated cervix in speculum examination, the direct biopsy must be performed for diagnosis. In this situation, the conization is not recommended and a biopsy is enough. Additionally, in this scenario, it is mandatory to perform a vaginal and rectal examination.

On the other hand, if patients have abnormal cytology, and/or positive results for DNA-HPV and no visible lesion in the cervix, which represents most of the early-stage diagnoses, they should undergo colposcopy and biopsies of suspicious areas. In this context, cytology has an important role in guiding the investigation; the slide must contain specimens from the squamocolumnar junction, endocervix, and ectocervix cells for evaluation. The colposcopy takes place by showing images that suggest tumoral invasion (e.g., atypical vessels, necrosis, or erosion) to guide the biopsy. During colposcopy, it is also important to access the endocervix by curettage. The endocervical curettage method can lead to up to 75% false-negative results; thus, the positive results should be considered, and a negative result does not exclude the possibility of cancer [18].

Finally, a cone biopsy is recommended when it is not possible to eliminate or estimate stromal invasion through the colposcopy and direct biopsy, or when the colposcopy is unsatisfactory, or cytopathological evaluation shows an intra-epithelial high-grade lesion, or even when there is a disagreement regarding methods for a suspect lesion. The cone biopsy allows the evaluation of histological type, maximum stromal invasion, tumor extension, and the presence or not of LVSI.

Regarding histopathologic diagnosis, the CC type can be classified as squamous cell carcinoma (keratinizing, non-keratinizing, papillary, basaloid, warty, verrucous, squamotransitional, or lymphoepithelioma-like), adenocarcinoma (endocervical, mucinous, villoglandular, or endometrioid), clear cell adenocarcinoma, serous carcinoma, adenosquamous carcinoma, glassy cell carcinoma, adenoid cystic carcinoma, adenoid basal carcinoma, small cell carcinoma, or undifferentiated carcinoma [19]. Since the squamous, adenocarcinoma, and adenosquamous types represent almost 100% of the diagnosed CC [20], all the discussions in this review will be conducted considering the literature for these three types.

## 3. Staging

After diagnosis, accurate staging is relevant for patients’ treatment plans and prognosis. Understanding the natural history of the disease is a key point to establishing staging systems. In this sense, it is known that CC can spread by extension into the vagina, parametrial tissue, uterus, bladder, or rectum; it also spreads to regional (pelvic) and para-aortic lymph nodes, and finally, distant metastasis can occur by the hematogenous route [21]. Therefore, the stagging system used worldwide is the International Federation of Gynecology and Obstetrician (FIGO), which determines the stage clinically, based on the tumor size and degree of pelvic extension, as represented in Table 1 [22]. In the last review published by FIGO in 2018, the CC stagging now includes pelvic and para-aortic lymph node status. In addition, imaging (US-Ultrasound, MRI-Magnetic Resonance Image, CT-Computerized Tomography, or PET- Positron Emission Tomography) was included as a complementary tool [23]. Another important update was that, with these changes, the FIGO and TNM (Tumor, Node, Metastasis) staging systems are now equivalent [24].

The decision of maintaining the clinical evaluation as the most important factor in staging was based on the knowledge that the highest rates of CC are reported in low-income countries where imaging can be challenging [21]. However, it is also known that staging based only on clinical findings is reliable just for patients placed in the extremes, such as IIIA, IIIB, or IVA [25]. Because of that, including lymph node evaluation and the possibility of imaging allows high-income settings to perform more accurate staging and highlights the relevance of lymph node assessment for oncologic outcomes and the concern in establishing less invasive approaches to evaluate them, two themes that have been widely discussed in the literature in the past 10 years [21].

Regarding imaging management, the current staging system allows the use of any imaging modality [23]. However, MRI has been considered the best method for primary tumor assessment [25,26]. On the other hand, for lymph node involvement evaluation, PET arises as to the best option since the publication of a meta-analysis including 5042 patients in which PET sensitivity and specificity rates were demonstrated to be superior when compared to the same rates for MRI and CT [27]. 

Regarding pathological findings, when a surgical specimen is available, pathologic staging should be performed, as it is a precise method to assess the extent of the disease [23]. Finally, the staging should be established at diagnosis and cannot be altered, even at recurrence; thus, it is recommended to close it after evaluating all available resources [23].

## 4. Treatment

The treatment options for early-stage CC are radiotherapy and surgery since both have the same rate of success in oncological outcomes [15]. A unique clinical randomized trial comparing primary surgery with primary radiotherapy in IB-IIA stages was conducted by Landoni et al. in 1997, showing that disease free-survival and overall survival for both groups were the same [15]. Nevertheless, surgery is the primary choice rather than radiotherapy because of quality-of-life issues and ovarian failure, the usual consequences of radiotherapy [28,29,30]. Regarding life quality, the data are still controversial but tend to show more complications and morbidity with radiotherapy [29]. Therefore, for young patients in general—with no comorbidities and when preservation of hormonal and sexual functions is relevant—surgery is the best option. In contrast, for elderly patients for whom the preservation of hormonal and sexual functions are not priorities compared to the risks of radical surgery, or for patients who are not candidates for surgery due to comorbidities or low functional status, radiotherapy with or without chemotherapy is recommended [21].

When surgery is chosen as the best approach considering the criteria above, RH is the standard of care [13]. However, the desire and possibility of fertility-sparing management by trachelectomy can be considered, depending on the stage of the disease and other risk factors for recurrence, such as tumor size and LVSI [31]. When RH is the selected option, other modalities of more conservative hysterectomy can be considered to avoid long-term sequelae depending on stage and risk factors such as lymph node status [23]. 

Risk factors in CC are classified as high, intermediate, or low. A risk factor is considered high when it includes positive surgical margins, parametrial invasion, and lymph node metastasis (Peters Criteria) [32]. Intermediate risk includes: (1) LVSI plus deep one-third cervical stromal invasion and tumor of any size; (2) the presence of LVSI plus middle one-third stromal invasion and tumor size > 2 cm; (3) the presence of LVSI plus superficial one-third stromal invasion and tumor size > 5 cm; and (4) no LVSI but deep or middle one-third stromal invasion and tumor size > 4 cm (Sedlis Criteria) [33]. Low-risk factors, although they do not have established specific criteria, are being reported in the literature as tumor size < 2 cm, no LVSI, depth of invasion < 10 mm, and no lymph-node involvement [34].

Finally, the rationale for early-stage CC treatment is presented in Figure 1 and the details of each topic will be discussed in the following subsections. 

### 4.1. Treatment for IA1 Stage

Patients in IA1 should be diagnosed through conization to establish deepness, tumoral extension, margins, and LVSI. If no LVSI is detected, there is a risk of 1% lymph node dissemination and recurrence [35]. Therefore, these patients can be treated conservatively, by conization with free margins or an extrafascial hysterectomy if they do not wish to preserve fertility [36]. The role of LVSI in the IA1 stage remains a controversial topic. However, most centers recommend RH or radiotherapy if this pathological finding is present [37].

### 4.2. Treatment for IA2, IB1, IB2, and IIA1 Stages

The standard recommendation for patients with early-stage cervical cancer (FIGO 2018 IA1 with LVSI-IIA1) remains the RH with lymph node evaluation, such as sentinel lymph node mapping and/or pelvic lymphadenectomy, with an overall survival between 70% and 90% [13]. 

For IA2 and IB1 stages, modified RH (vaginal margins 1–2 cm and cardinal ligaments divided where ureter transits parametrial tissue, while on RH, the vaginal margin is, at minimum, one-quarter and cardinal ligaments are divided at the pelvic sidewall) can be considered [38]. The possibility of less radical hysterectomy for these stages is due to the low risk of parametrial invasion [38] since the morbidity related to the RH is mostly because of the parametrectomy. The damage to autonomic nerve fibers from the bladder, bowel, and sexual organs during parametrial removal is responsible for morbidity; thus, it is reasonable to perform more conservative surgery when it is oncologically safe [39]. 

In this regard, Frumovitz et al. evaluated the incidence of parametrial enrollment in women with early-stage CC that were submitted to RH. Patients presenting squamous cell carcinoma, adenocarcinoma, or adenosquamous carcinoma, at stages IA2-IB1, were enrolled. Considering all patients (*n* = 350), just 7.7% presented parametrial invasion. Evaluating only the ones with tumors < 2 cm and no LVSI, there was no histological evidence of parametrial involvement in any patient [40]. Additionally, the ConCerv study, recently published, advocated in favor of more conservative surgery in stages IA2 and IB1 with a low-risk profile. They evaluated 100 patients, and the reported rate of positive lymph nodes was 5%, the rate of residual disease in the hysterectomy specimen followed by conization was 2.5%, and the 2-year overall recurrence rate was only 3.5% [41].

Despite the promising results discussed above, we should wait for the publication of two clinical trials—SHAPE (NCT03705650) and GOG 278 (NCT01649089)—that are evaluating the oncologic safety of less radical surgery in patients with CC with tumors that are up to 2 cm in order to obtain more solid evidence about the role of conservative surgery for this group. 

### 4.3. Lymph Node Staging

For years, lymph node involvement in CC has been recognized as a crucial parameter for therapeutic decisions and prognosis [42]. Traditionally, lymph node staging is per-formed by lymphadenectomy during RH. However, in recent years, less morbid options—such as imaging and sentinel lymph node (SLN) biopsy—have emerged with feasible results [43,44]. 

As for imaging, PET is considered the best option for lymph node assessment due to its higher sensitivity and specificity (73% and 98%, respectively) compared to MRI (56% and 93%) and CT (58% and 92%) [27]. The increase in sensitivity appears to be due to the improved ability of PET to detect abdominal lymph node metastases, which may alter radiotherapy planning and prognosis [45].

SLN biopsy is increasingly used as a substitute for systematic pelvic lymphadenectomy due to surgery-related morbidity, particularly lower limb lymphedema [46]. However, it cannot be used in all cases of CC because of four criteria for selecting patients who can undergo SLN biopsy: tumors < 4 cm; no suspicious lymph nodes on imaging; bilateral evidence of SNL; and availability of ultrastaging (advanced pathologic review) [47]. Based on these criteria, most early-stage cases may benefit from this technique.

In this context, SLN mapping came into being. It is performed by injecting a dye (such as patent blue or indocyanine green) or radiopharmaceutical substances (such as technetium 99) into the cervix. Nowadays, indocyanine green is the best option for isolated use because of its high detection rates and rare side effects such as allergies [48]. If the anatomical pathological examination of the frozen section reveals positive findings for metastases, most health centers will cancel the radical surgery and perform chemoradiotherapy [49].

At this point, it is important to notice that the indication of the frozen section technique is not a consensus, as it presents accuracy limitations. Slama et al. [50], in the largest cohort of SLN assessment by frozen section, found high rates of false-negative results, mainly due to the technical inability to detect low-volume metastases as the congelation process can spoil the ultrastaging. However, the European guidelines [51] recommend the use of frozen sectioning as it can be performed intraoperatively—helping to decide whether to proceed with RH or avoid combined treatment—even though the value of micrometastases to treatment and prognosis is unknown.

Therefore, the results of SLN mapping are promising so far. In retrospective series, false-negative rates are less than 1% [21]. Moreover, in a recent meta-analysis, the prevalence of SLN metastases for CC early stage was 21%, the sensibility was 94%, with a negative predictive value between 91% and 100% and a false-negative rate of 1.5% [52]. However, we should wait for the results of clinical trials such as SENTIX, PHENIX, and SENTICOL III to provide answers to the question of whether systematic pelvic lymphadenectomy can be safely replaced by SLN mapping.

SNL biopsy also allows the search for low-volume metastases [46]. This search is performed by ultrastaging, a more detailed pathological analysis of the SNL that must include, at a minimum, additional sections, staining with H&E, and immunohistochemistry [53]. This technique increases the identification of lymph node metastases by 15%, as it also detects low-volume metastases [54]. These metastases are divided into micrometastases (MIC, tumor deposit between 0.2 and 2 mm) and isolated tumor cells (ITCs, tumor deposit < 0.2 mm), while macrometastases (MAC) are considered as tumor deposits > 2 mm [55]. Although the clinical relevance of MAC for the indication of adjuvant treatment due to poorer prognosis is a consensus, the impact of detecting MIC or ICTs remains unknown.

Several studies investigated this question retrospectively, with heterogeneous results. To our knowledge, the largest study in this regard was conducted by Cibula et al. [54], which included 645 patients and showed an association between MAC or MIC with a reduction in overall survival. Moreover, Marchiolé et al. [56] found that the presence of MIC was an independent risk factor for recurrence. On the other hand, Zaal et al. [57] found that in the presence of MIC, overall survival was better when more than 16 lymph nodes were dissected. No prognostic relevance has yet been found for ITCs [58]. However, it is difficult to compare these studies because they differ in their methodology, especially regarding the ultrastaging protocol used and the selection of patients. 

In this sense, Guani et al. [59] presented a prospective evaluation of recurrence and survival in CC early-stage patients with MIC or ITCs. No effect of the presence of MIC or ITCs on progression-free survival was found. However, the authors emphasized that although the prospective nature of the study was an advantage, the number of patients included (139) was insufficient to answer the question; thus, the results could not be considered definitive.

In conclusion, further research is needed to determine the true impact of low-volume metastases on disease progression. However, in a survey conducted by ESGO in 2018 [60], it was found that 93% of practitioners consider MIC as a parameter to indicate adjuvant therapy, but not ITCs.

### 4.4. Fertility Sparing-Surgery

Data from SEER show that approximately 42% of women diagnosed with CC are younger than 45 years of age [61]. Due to this high incidence in young patients, who often desire to preserve their fertility, the techniques of fertility-sparing surgery have emerged as a demand on surgeons and hope for these patients. Trachelectomy consists of removal of the entire cervix, parametrial tissue, and vaginal cuff, preserving the uterus (body and fundus), ovaries, and tubes, with SLN mapping with or without bilateral pelvic lymphadenectomy [62]. 

Radical trachelectomy was initially described using only a vaginal approach. In the last decade, the open approach began to be used, and today both approaches, vaginal and abdominal, are acceptable because the results are comparable in terms of complications [63].

The indications for trachelectomy are: 40 years old or younger, intends to have children, fertile patient, stage IA1 with LVSI or IA2 and IB1 stages (tumor size < 2 cm) with MRI not showing parametrial invasion or metastases to lymph nodes or other sites. Considering this, about 50% of patients with CC under 40 years of age may be candidates for this procedure [31]. 

Regarding oncologic safety, there was no difference in cure rate between radical trachelectomy and RH. In a case-control observational study comparing radical trachelectomy (vaginal or abdominal) and RH for tumors up to 4 cm, recurrence-free survival and disease-specific survival were similar, indicating that oncologic outcomes can be considered equivalent between the procedures [64]. 

Regarding reproductive function after radical trachelectomy, patients who attempt to conceive have a 50% chance of being successful, and in those who do, the risk of miscarriage in the first trimester is the same as in the general population, approximately 20%, although this risk increases in the second trimester [63,65]. In a study that included 200 pregnancies after trachelectomy, 75% of these ended in live births [66].

A remaining question is the oncologic safety of radical trachelectomy, whether performed openly or minimally invasively. The recently published IRTA study [67] retrospectively addressed this issue and evaluated 646 patients with early-stage tumor CC up to 2 cm in size for 4.5-year disease-free survival, overall survival, and recurrence rate. No difference was found when comparing the two groups for these three oncologic outcomes. However, the authors pointed out the limitations of the study—such as the retrospective nature and the low recurrence rates in the group of patients studied—and emphasized that further studies are needed [67].

More recently, another option for fertility-preserving treatment was presented in a group of patients not captured by the trachelectomy selection criteria. Vincenzo et al. [68] studied neoadjuvant chemotherapy followed by conization in young IB2-IIA1 patients with unfavorable pathological aspects not eligible for trachelectomy, such as tumor size between 2 and 4 cm and LVSI, who wished to preserve their fertility. Regarding oncologic outcomes, the authors found that there was no difference in the overall survival rate between that reported after using their protocol (90.9%) and that reported in the literature for RH (91%) [69]. Regarding obstetric outcomes, the pregnancy rate in their series was 66.7%, which is comparable to the literature for trachelectomy. However, the authors pointed out that the small number of patients included (13 in the final analysis) was a limitation, although the study was prospective and the patient selection and follow-up time were adequate. Therefore, this topic is a good area for further investigation in larger clinical trials.

### 4.5. Adjuvant Treatment

The decision to implement adjuvant treatment in early-stage CC patients is based on the risk of disease recurrence, as recommended for intermediate and high-risk patients [70]. The criteria of Sedlis and Peters mentioned earlier in this section [32,33] are used to classify patients as intermediate or high risk.

In intermediate-risk patients, the risk of recurrence and death after surgery alone is approximately 30% [33,70]. In high-risk patients, the risk of recurrence and death after surgery alone increases to 40–50% [32,71], so adjuvant therapy is indicated in both cases. Chemoradiation seems to be the best option in high-risk patients, as in a retrospective analysis [72], the recurrence rate was lower in the chemoradiation group and, thus, the five-year progression-free survival was better than in the group receiving radiotherapy alone. However, radiotherapy alone is superior to no treatment after surgery in this context, as a meta-analysis observed a reduction in the risk of recurrence for the group that received radiation compared to the group without further therapy [73].

In this sense, however, the difficulty remains in identifying patients who require adjuvant therapy before surgery. Since the previously used criteria for intermediate and high risk depend on pathological aspects, the only option nowadays is a postoperative evaluation, regardless of the surgical approach chosen [74]. In this field, tumor-free distance (TFD) proves to be a relevant option, as it can be assessed by MRI before surgery. Bizzarri et al. [75] validated TFD as a prognostic marker that could help in the decision to undertake adjuvant therapy. The cut-off TFD value used by the authors was 3 mm, and they found that a TFD ≤ 3 mm was a strong predictor of lymph node involvement and was associated with worse 5-year disease-free survival and overall survival. In addition, there was concordance between TFD measurement on MRI and histology after surgery.

## 5. Surgical Approach

Since surgery was established as primary care in early-stage CC, the surgical approach, open surgery (OS or laparotomy) or minimally invasive surgery (MIS—laparoscopic or robotic) became an issue of interest. The first laparoscopic RH with pelvic and paraaortic lymphadenectomy was performed by Nezhat and contributors in 1989 and published in the early nineties [76]. Since then, an increase in minimally invasive modalities, laparoscopic or robotic, was observed when performing the Rh. Several observational studies have demonstrated that the prognostics were similar between the laparotomic and minimally invasive approaches. When compared, the results showed that the widely adopted laparoscopic approach was associated with less blood loss during surgery and lower rates of postoperative complications [77]. However, these studies were retrospective and did not focus on oncologic outcomes as the main objectives.

In 2018, a randomized clinical trial was published comparing the disease-free survival between the laparotomic and the minimally invasive approaches’ RH [34]. The LACC Trial was multicentric, randomized, and controlled. It was the first study that prospectively compared the laparotomic RH with minimal invasive RH (laparoscopic or robotic). Moreover, the disease-free survival (DFS) and overall survival (OS) were evaluated between the groups. The results were unexpected, showing a lower rate of DFS and OS in the MIS arm [34]. The impact of this trial was beyond the surgical approach since they had collected data to address other issues such as adverse events and quality of life. Although MIS has been historically associated with better short-term quality of life and esthetic satisfaction, less blood loss, and decreased risk of adverse surgical events [78,79,80], using LACC trial data, Obermair et al. reported no difference in the overall incidence of intraoperative or postoperative adverse events between OS and MIS groups [81] and Frumovitz et al. also demonstrated no difference in quality-of-life scores between the groups [82].

Since the publication of this trial, several centers worldwide accessed their data retrospectively to investigate whether or not they could confirm the LACC trial results [83,84,85,86,87,88]. In most of them, the data corroborated the LACC trial findings. Melamed et al. compared patients with CC IA2 or IB1 (FIGO 2014) submitted to open versus minimally invasive RH from a cohort based on the SEER (Surveillance, Epidemiology and End Results) database. In this study, minimally invasive surgery was associated with an increase in the risk of death in 4 years in comparison to open surgery (9.1% vs. 5.3%). In addition, they observed that the adoption of minimally invasive surgery in the USA in 2006 was concomitant with the start of declining rates of overall survival in 4 years [88]. 

Because of the LACC trial and these other recent studies, the ESGO (European Society of Gynaecological Oncology) and the NCCN (National Comprehensive Cancer Network) guidelines were reviewed to formally recommend open surgery as the gold-standard method to perform a radical hysterectomy.

Another missing point on the LACC trial is the reason why performing MIS has poor oncologic outcomes. The possible explanations raised were the use of uterine manipulator and insufflation gas (CO₂) [34]. In this regard, the use of protective measures that could avoid these harmful effects of MIS, such as the vaginal protective maneuver and conization before surgery, were investigated since benefits were reported in terms of complications and quality-of-life [78,79,80]. Two retrospective studies [89,90] showed no difference between the OS and MIS groups’ oncologic outcomes when the vaginal protective maneuver was performed in MIS. Another two retrospective studies [91,92] observed a protective effect of conization before surgery. However, both studies had limitations due to their observational nature, and the number of patients enrolled or evaluated as a secondary endpoint.

## 6. Tumor Size < 2 cm

The LACC trial’s impact on the role of MIS for CC treatment is undeniable since a decrease in MIS procedures was reported after its publication [93]. However, it also called into question the safety of MIS for patients with a low-risk profile because the study was not statistically powered to drive conclusions for this group of patients [34]. A low-risk profile patient is characterized by tumor size < 2 cm, no LVSI, depth of invasion < 10 mm, and no lymph-node involvement [34].

In this sense, tumor size above 2 cm, as illustrated in Figure 2, was reported to be the only factor that independently has prognostic impact depending on the surgical approach [94]. Because of this, in recent years, several retrospective studies addressed the oncologic safety of MIS for patients with tumor size < 2 cm. Despite these efforts, the question remains open; some of the studies showed no difference [90,94,95,96,97,98] and others demonstrated the association of MIS with poor oncologic outcomes [99,100,101,102], as summarized in Table 2.

Among the authors who reported no difference, Pedone Anchora et al. [94] and Rodriguez et al. [98] observed no difference in DFS rates between the MIS and open surgery groups. In addition, Nam et al. [95] and Chiva et al. [90] evaluated recurrence-free survival and risk of recurrence, respectively, showing no difference too. Lastly, Yang et al. [97] reported no difference in OS rates.

On the other hand, among the authors who reported differences between the groups, Paik et al. and Chen et al. reported lower DFS in the MIS group compared to the open approach (HR 12.987, 95% CI 1.451 to 116.244, and HR 4.64, 95% CI 1.26 to 17.06, respectively) [101,103]. Additionally, Uppal et al. reported lower DFS (HR 2.83; 95% CI, 1.1 to 7.18) and an increased hazard for recurrence (HR, 1.88; 95% CI, 1.04 to 3.25) in the MIS group [99]. Lastly, Nasioudis et al. reported worse overall survival in the MIS group (HR 1.72; 95% confidence interval, 1.05 to 2.82) [102].

The disagreements are possibly related to the retrospective nature of these studies, mainly due to the insufficient number of participants, low accuracy of determination of tumor size, and lack of control of confounder variables such as histology, LVSI, and post-operative treatment [104]. Regarding the assessment of tumor size, data comparing size determination by clinical evaluation with pathology findings demonstrated high variability, probably inducing inadequate patient selection [99].

Other studies are required to obtain a reliable answer regarding this issue, especially prospective ones, in which the variables can be well controlled, and the number of participants powered for statistical analysis.

## 7. Prognosis

Although early-stage CC has a good prognosis with an overall survival rate of 70–90% [13], relapses and deaths are to be expected. Therefore, it is important to know the factors that influence the poor course of the disease in order to choose between more conservative or aggressive treatment and to help patients define and weigh their wishes and expectations during treatment. 

Today, the factors that are most important for prognosis in CC are staging, tumor size, lymph node involvement, depth of stromal invasion, and LVSI [105]. Obviously, staging is the most clinically relevant as it takes into account several aspects. However, as these factors are interrelated, it can be challenging to determine the most important factor, or at least the ones that are independently important. In this context, the studies in which multivariate regression analysis has been performed make an important contribution and are therefore discussed below, although they still provide conflicting results.

After staging, lymph node involvement is considered the second most important aspect for prognosis, as the five-year survival rate for patients with early-stage CC without lymph node involvement is about 90%, while it drops to 60.8% for the same group but with lymph node involvement [106]. Due to these enormous effects, lymph node status is one of the criteria for the indication of adjuvant treatment. 

Regarding tumor size, both Wright et al [106] and Wagner et al [107] found that tumor size is the most important prognostic factor, as it is related to parametrium and lymph node involvement and decreases survival rates. This was reflected in a better survival rate in women with lymph node involvement and smaller tumor size compared to women without lymph node involvement but with tumors > 4cm.

Regarding LVSI, Rutledge et al [108] and Creasman and Kohler [109] presented divergent data. While the first study showed that prognosis seems to be most influenced by the presence of LVSI rather than tumor size, as suggested by the staging system, the second study showed that LVSI is not an independent risk factor.

## 8. Conclusions

The treatment for CC in early stages, up to 4 cm, has been evolving in recent years. The current scenario indicates surgery as the standard of care, especially when conducted following the laparotomy approach, for which superiority is demonstrated. Moreover, trials are being conducted to evaluate both the necessity of parametrectomy in patients with tumors up to 2 cm and the role of SLN mapping in patients with early-stage CC. These advances show pathways for more tailored treatment, avoiding morbidity while cancer control is assured. Nowadays, in high-income countries, the patients diagnosed with CC receive the first-line recommended treatments discussed here, with access to imaging, radiotherapy, chemotherapy, and more conservative surgical possibilities covered. A lot of effort should be made to make all these advances available in low-income countries in the coming years, as they represent 85% of CC cases worldwide.

Nonetheless, prevention remains the best option to achieve substantial declines in the rates of mortality and morbidity by CC. Thus, as well as the improvement of treatment, finance and the encouragement of effective prevention measures such as HPV vaccination and screening should be an urgent goal since almost all CC cases have high-risk HPV as the main cause.

## Figures and Tables

**Figure 1 cancers-14-00575-f001:**
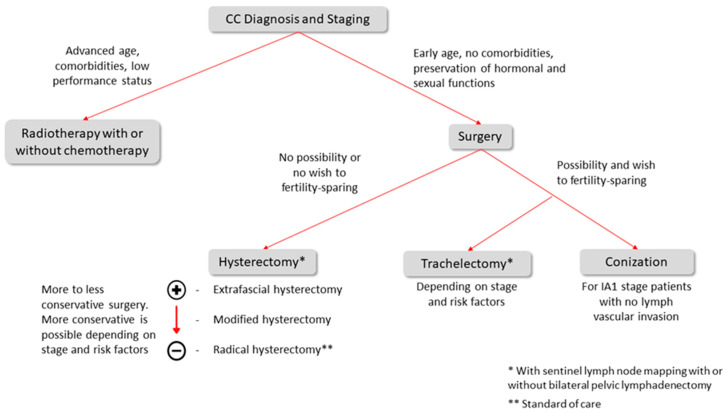
Early-stage CC treatment rationale.

**Figure 2 cancers-14-00575-f002:**
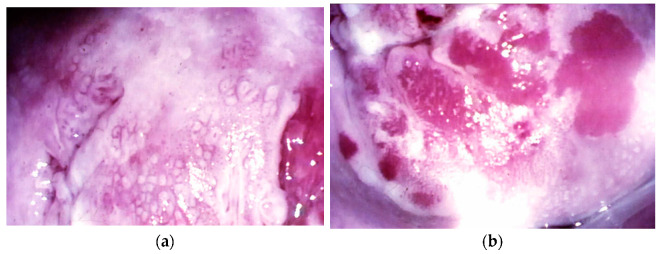
Tumors < 2 cm. (**a**) A 14 × resolution image of a microinvasive lesion on colposcopy evaluation; (**b**) A 14 × resolution image of a stage IA2 squamous tumor: colposcopy evaluation of atypical vessels.

**Table 1 cancers-14-00575-t001:** FIGO Stage System.

Stage	Description
I	The carcinoma is strictly confined to the cervix (extension to the uterine corpus should be disregarded)
IA	Invasive carcinoma that can be diagnosed only by microscopy, with maximum depth of invasion ≤ 5 mm ^a^
IA1	Measured stromal invasion ≤ 3 mm in depth
IA2	Measured stromal invasion > 3 mm and ≤5 mm in depth
IB	Invasive carcinoma with measured deepest invasion > 5 mm (greater than Stage IA); lesion limited to the cervix uteri with size measure by maximum tumor diameter ^b^
IB1	Invasive carcinoma > 5 mm depth of stromal invasion, and ≤2 cm in greatest dimension
IB2	Invasive carcinoma > 2 cm and ≤4 cm in greatest dimension
IB3	Invasive carcinoma > 4 cm in greatest dimension
II	The cervical carcinoma has invaded beyond the uterus, but has not extended onto the lower third of the vagina or to the pelvic wall
IIA	Involvement limited to the upper two-thirds of the vagina without parametrial invasion
IIA1	Invasive carcinoma ≤ 4 cm in greatest dimension
IIA2	Invasive carcinoma > 4 cm in greatest dimension
IIB	With parametrial invasion but not up to the pelvic wall
III	The carcinoma involves the lower third of the vagina and/or extends to the pelvic wall and/or causes hydronephrosis or non-functioning kidney and/or involves pelvic and/or paraaortic lymph nodes ^c^
IIIA	Carcinoma involves lower third of the vagina, with no extension to the pelvic wall
IIIB	Extension to the pelvic wall and/or hydronephrosis or non-functioning kidney (unless known to be due to another cause)
IIIC	Involvement of pelvic and/or paraaortic lymph nodes (including micrometastasis) ^c^, irrespective of tumor size and extent (with r and p notations) ^d^
IIIC1	Pelvic lymph node metastasis only
IIIC2	Paraaortic lymph node metastasis
IV	The carcinoma has extended beyond the true pelvis or has involved (biopsy proven) the mucosa of the bladder or rectum. A bullous edema, as such, does not permit a case to be allotted to Stage IV
IVA	Spread of the growth to adjacent organs
IVB	Spread to distant organs

^a^ Imaging and pathology can be used, when available, to supplement clinical findings with respect to tumor size and extent, in all stages. ^b^ The involvement of vascular/lymphatic spaces should not change the staging. The lateral extent of the lesion is no longer considered. ^c^ Isolated tumor cells do not change the stage but their presence should be recorded. ^d^ The addition of the notation of r (imaging) and p (pathology) to indicate the findings that are used to allocate the case to Stage IIIC. For example, if imaging indicates pelvic lymph node metastasis, the stage allocation would be Stage IIIC1r; if confirmed by pathological findings, it would be Stage IIIC1p. The type of imaging modality of pathology technique used should always be documented. When in doubt, the lower staging should be assigned. Source: Corrigendum to “Revised FIGO staging for carcinoma of the cervix uteri” [Int J Gynecol Obstet 145(2019) 129-135] [22].

**Table 2 cancers-14-00575-t002:** Characteristics of the studies evaluating oncological outcomes in patients with tumors < 2 cm.

Author	Year	N	Outcomes
Nam, et al. [95]	2012	526 (335 < 2 cm)	No difference between open surgery (OP) and minimally invasive surgery (MIS) for oncologic outcomes
Paik, et al. [101]	2019	476 (248 < 2 cm)	Difference observed: MIS was associated with a lower rate of disease-free survival (DFS)
Kim, et al. [96]	2019	565 (283 < 2 cm)	No difference between open surgery (OP) and minimally invasive surgery (MIS) for oncologic outcomes
Pedone Anchora, et al. [94]	2020	423 (251 < 2 cm)	No difference between open surgery (OP) and minimally invasive surgery (MIS) for oncologic outcomes
Chen, et al. [103]	2020	325	Difference observed: MIS was associated with worse 5-year disease-free survival
Yang, et al. [97]	2020	333 (111 < 2 cm)	No difference between open surgery (OP) and minimally invasive surgery (MIS) for oncologic outcomes
Chiva, et al. [90]	2020	693 (303 < 2 cm)	No difference between open surgery (OP) and minimally invasive surgery (MIS) for oncologic outcomes
Uppal, et al. [99]	2020	815 (264 < 2 cm)	Difference observed: MIS was associated with increased risk of recurrence and inferior disease-free survival
Rodriguez, et al. [98]	2021	1379 (979 < 2 cm)	No difference between open surgery (OP) and minimally invasive surgery (MIS) for oncologic outcomes
Nasioudis, et al. [102]	2021	2046	Difference observed: MIS was associated with worse overall survival (OS)

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
