# Peer review of "Management of Early-Stage Cervical Cancer: A Literature Review"

_cancers, 2022, doi:10.3390/cancers14030575_

Round 1

Reviewer 1 Report

I think the authors have adequately revised the manuscript according to suggestions by reviewers. I appreciate having a chance to review your excellent article on early-stage cervical cancer.

This manuscript is a resubmission of an earlier submission. The following is a list of the peer review reports and author responses from that submission.

Round 1

Reviewer 1

  1. In this review article, authors discuss the advances and ongoing clinical trials in surgical treatment in the literature search. The composition of the sections included diagnosis, staging, treatment, surgical approach, and tumor size <=2cm. The treatment section was subdivided into treatment for IA1 stage, stage IA2, IB1, IB2, and IIA1 stage, sentinel lymph node mapping, and fertility-sparing

The manuscript is well written based on the generally accepted discussions and consensus. The selected topics are valid but seem to lack a section of prognostic factors. Is it possible to include discussions on prognostic factors?

Answer: Thank you for your suggestion, we added a new section to discuss this topic (“7. Prognosis”).

Minor point:

  1. Please correct the word “Traquelectomy” to “Trachelectomy” in Figure

Answer: Thank you for your correction, we corrected it.

    3. Please check if the abbreviations for the journal's title in the Reference are correct, g., Journal of the Advanced Practitioner in Oncology, Gynecologic Oncology, and so on. OK

Answer: Thank you for your suggestion, we checked the abbreviations and it seems to be okay,

as it was automatically generated by EndNote software when exported from PubMed.

Reviewer 2

Thank you for the interesting article. I have few comments:

  1. The simple summary and abstract show the same Please show more results in the abstract.

Answer: Thank you for your suggestion, we changed the abstract section.

  1. In Figure 1 (line 198), I think lymphatic staging should be added.

Answer: Thank you for your excellent suggestion, we included the information about lymphatic staging in Figure 1 at Hysterectomy and Trachelectomy.

  1. Ch 3 line 236-254: I should title this chapter "Lymph nodal staging". Can you discuss here the frozen section and treatment of micrometastasis?

Answer: Thank you for your suggestion, we have included information about the sentinel frozen section and management of micrometastasis at subsection “4.3 Lymph Nodal Staging”.

  1. Line 252: please, add the other prospective trials, SentiX and

Answer: Thank you for your suggestion, we have included this information on line 285.